A domain-specific language for managing ETL processes

Popović Aleksandar 1 aleksandarp@ucg.ac.me
Ivković Vladimir 2
Trajković Nikola 3
http://orcid.org/0000-0003-1319-488X Luković Ivan 4 ivan.lukovic@fon.bg.ac.rs
1 Faculty of Science and Mathematics, University of Montenegro , Podgorica , Montenegro
2 Faculty of Technical Sciences, University of Novi Sad , Novi Sad , Serbia
3 Softwecs LTD , Podgorica , Montenegro
4 Faculty of Organizational Sciences, University of Belgrade , Belgrade , Serbia
Maguitman Ana
Electronic publication date: 2024 Jan 26
Publication date: 2024
Volume: 10
Electronic Location ID: e1835
Received 2023 May 8; Accepted 2024 Jan 3
Copyright: © 2024 Popović et al.
Copyright year: 2024
Copyright holder: Popović et al.
License: This is an open access article distributed under the terms of the Creative Commons Attribution License, which permits unrestricted use, distribution, reproduction and adaptation in any medium and for any purpose provided that it is properly attributed. For attribution, the original author(s), title, publication source (PeerJ Computer Science) and either DOI or URL of the article must be cited.
License URL: https://creativecommons.org/licenses/by/4.0/

Keywords: Domain-specific language, Extraction transformation and loading, Data warehouse, Platform-independent models, Model-driven development

Funding: The authors received no funding for this work.

==============================
Maintenance of Data Warehouse (DW) systems is a critical task because any downtime or data loss can have significant consequences on business applications. Existing DW maintenance solutions mostly rely on concrete technologies and tools that are dependent on: the platform on which the DW system was created; the specific data extraction, transformation, and loading (ETL) tool; and the database language the DW uses. Different languages for different versions of DW systems make organizing DW processes difficult, as minimal changes in the structure require major changes in the application code for managing ETL processes. This article proposes a domain-specific language (DSL) for ETL process management that mitigates these problems by centralizing all program logic, making it independent from a particular platform. This approach would simplify DW system maintenance. The platform-independent language proposed in this article also provides an easier way to create a unified environment to control DW processes, regardless of the language, environment, or ETL tool the DW uses.

Introduction

In a highly competitive world, managers need consistent and accurate data to make informed decisions for the benefit of their organizations. This demand for high-quality decision support data has initiated the development of new approaches and methodologies for organizing and structuring data, including data ingestion from diverse sources, historical data storage, search functionality, analytical processing, and reporting. A predominant approach in this domain relies on the Data Warehouse (DW) paradigm (Kimball & Ross, 2013). A DW is a copy of transactional data specifically structured for queries and analytical processing (Inmon, 2005). The DW structure consists of data and associated data manipulation mechanisms. These mechanisms include processes aimed at data extraction, transformation, and loading (ETL). The design and implementation of ETL processes is often the most demanding task of the DW development process, and substantial efforts have been made to develop various tools that automate the ETL process.

Various ETL tools are available that support the extraction of data extraction from various heterogeneous sources, and the performance of diverse tasks such as initial loading, historical data loading from legacy systems, and incremental loading. Each tool has a specific structure and a list of supported functions that are related to the concrete technology for which the tool was built. ETL process design depends on both a selected ETL tool and a database language used for data manipulation. Batch scripts for running ETL tasks depend on a server operating system. Technological diversity and complexity hinder the orchestration and maintenance of the ETL processes being designed.

A focus of the research presented in this article is set to software companies specialized in the design and implementation of ETL processes. As a rule, such companies create hundreds of ETL processes using different tools and deploy them on different execution platforms. An ETL process is defined as a sequence of well-defined tasks where each task has a clear responsibility. Some tasks are common to many ETL processes, even though they are initially built for different purposes. Therefore, multiple ETL processes practically share the same or similar tasks. The majority of modern ETL tools are capable of creating reusable components. Because software companies develop and deliver their DW services to different software platforms, these services use multiple ETL tools for a single ETL process. If a single ETL process needs to extract data from multiple sources and no single ETL tool supports data extraction from all the sources, then multiple ETL tools are needed. Thus, a systematic approach is needed to help ETL designers establish and maintain a repository of reusable tasks and high-level specifications of new and existing ETL processes.

There are currently no tools on the market that enable platform-independent modeling of ETL orchestration tasks that can execute on heterogeneous platforms. Data engineers mostly use concrete ETL tools to perform daily tasks, but these tools do not provide simultaneous generation of executable specifications, such as SQL scripts or batch scripts, on more than one platform. One research goal of this article was to create a new domain-specific language (DSL) to facilitate the orchestration and maintenance of ETL processes at the level of platform-independent models (PIMs). This language was named the ETL Control Language (ETLCL), and it allows a user to create specifications for controlling data flow and loading data into DWs, regardless of platform or ETL tool. By this, our goal is to raise the level of abstraction when it comes to controlling ETL processes in DW systems.

Many organizations migrate data from complex operational databases into DWs on a daily basis, so there is a strong demand for an efficient, safe, and comprehensive environment for controlling ETL processes. Another goal of the ETLCL language is to facilitate error detection and handling by shifting focus from concrete technology and tools to the specifications created by high-level PIM concepts.

The concepts of the ETLCL language are abstractions of concepts and commands commonly encountered in tools and platforms widely used in the DW domain, providing a foundation for specifications in this language to be executable on various technological platforms. The main commands of the ETLCL language are related to task organization, such as defining tasks, defining dependencies between tasks, and scheduling task execution. Additionally, tasks can be grouped, and dependencies between groups can be defined.

To provide a fully functional DSL for designing ETL processes, ETLCL programs can also be transformed into executable code for a selected platform, such as a SQL script for a target DBMS, such as Microsoft SQL Server and Oracle DBMS, or a batch script for a target ETL tool, such as Oracle Data Integrator (ODI) or SQL Server Integration Services (SSIS) and Informatica for Linux and Windows platforms.

Dealing with ETL processes is not an easy task, and it requires expert knowledge from several fields, such as database systems and system administration. The goal of this article is to provide a simple, yet powerful language that enables users without expertise in the field to handle ETL processes. This platform-independent language is also a powerful tool for time-consuming tasks such as reorganizing DWs or partially reloading data into storage.

Defining and maintaining ETL processes is a highly error-prone task, even for experienced and skilled professionals. Using ETLCL as a PIM language that provides high-level concepts and commands can help mitigate the error risks. Also, our approach facilitates portability. Introducing a new platform, target language, or ETL tool does not require the complete development of new specifications for ETL processes. ETLCL only requires amendments of existing transformation algorithms for these changes. Deploying the ETLCL in this context provides additional benefits, including improved expressiveness, as discussed in Mernik, Heering & Sloane (2005).

Apart from the introduction and conclusion, this article is organized into five sections: “Research and Engineering Methodology” presents the methodology used in the language development, “Related Work” presents related work, “Main Concepts of the ETLCL Language” is devoted to the main concepts of the ETLCL language, “Application of ETLCL in a Use Case” presents a selected case study to demonstrate a typical application of ETLCL, and “An Assessment of ETLCL’s Main Characteristics” evaluates the ETLCL language through a discussion and assessment of its characteristics.

Research and engineering methodology

One of the crucial steps in language development is language design, which involves gathering and analyzing requirements, defining key concepts, specifying abstract syntax, and choosing appropriate notations. Karagiannis (2018) introduced the Agile Modeling Method Engineering (AMME) approach that combines agile principles with modeling techniques, enabling the rapid and iterative development of modeling methods tailored to specific projects or organizational needs. Karagiannis et al. (2019) further explore modeling method requirements, and as a result the authors introduced CoChaCo (Concept-Characteristic-Connector) method in order to appropriately support earlier stages in the AMME approach. Frank (2013) presented a methodological framework for language design that primarily concentrates on aiding the process of analyzing requirements. Numerous strategies and methodologies have been introduced to implement DSLs, including techniques like embedding, interpretation, preprocessing, and compiler/application generator approaches (Mernik, Heering & Sloane, 2005). For example, Domain-Specific Modeling (DSM) is a methodology with numerous successful applications, where the emphasis is placed on the implementation and use of graphical DSLs, as well as the generation of complete source code (Kelly & Tolvanen, 2005). In addition to the actual DSL implementation, a crucial step in language development is its evaluation. Over the last two decades, this topic has garnered significant attention from the academic community, leading to the development of approaches and frameworks for assessing DSLs. Kahraman & Bilgen (2013) introduced a comprehensive framework for evaluating language characteristics.

The rest of this section describes the principles and approach used to create the ETLCL language. The general principles provided in the previously-referenced publications were applied in the formulation, implementation, and evaluation of this DSL. In the design phase, we defined the most crucial requirements that the ETLCL language should meet. These requirements serve as a starting point for identifying the language characteristics to be evaluated in the later stages of development. The evaluation of this language and its characteristics is provided in “An Assessment of ETLCL’s Main Characteristics”.

Requirements and design

The first step in language implementation is the requirement analysis and language design phase. This step is demanding as the language designer must closely collaborate with domain experts to effectively express domain knowledge through the language concepts being designed. Lack of support from users and ambiguous requirements are some of the primary obstacles that can arise in the requirement gathering phase (Frank, 2013). This challenge was mitigated in the design process of the ETLCTL language as the designers and authors of this article possess both experience in language design and implementation, as well as practical and theoretical knowledge in the DW domain. The following are requirements for this phase:

R1—Language concepts must originate from the domain of orchestrating ETL activities and be familiar to the targeted group of end-users, i.e., ETL experts. The goal is to achieve a high degree of usability and to provide users with a language that is relatively easy to learn.

R2—Semantics of language concepts must be invariant within the scope of the target domain since it is very technically heterogeneous, as there are a number of ETL tools, DBMSs, and platforms in widespread use.

R3—Language concepts must be expressive enough to enable users to comprehensively describe the majority of typical scenarios related to orchestrating ETL processes in DW systems. The goal is to create a language that provides a high level of functional suitability.

R4—The DSL must exhibit high portability, allowing seamless deployment and execution across diverse computing environments, including various operating systems and ETL tools.

R5—The language must promote productivity; orchestrating a typical ETL process through the ETLCL language should take less time compared to orchestrating an equivalent process through a concrete environment and tools.

In order to meet requirements R1 and R2, an analysis of current ETL tools and DBMSs was performed, and a significant number of concepts were identified as being common in all of them. The main concepts and commands of ETLCL were created using an abstraction of concepts that are supported by the majority of tools and platforms used in practice. The Task concept is a central concept of the ETLCL language, representing an atomic operation within ETL activities, which is present in the majority of tools and platforms, and have common semantics. Among tasks, there is often a well-defined execution order based on dependencies. For example, the execution of one task may depend on the successful completion of a preceding task. Concepts such as Dependency, Group, and Scheduler were introduced in order to model such scenarios. Such a selection of concepts also lays the groundwork for satisfying requirement R4 related to portability, as the chosen concepts can be mapped and transformed into corresponding commands for a target computational environment. Figure 1 contains a meta-model of the ETLCL language, while “Main Concepts of the ETLCL Language” contains a detailed description of ETLCL concepts.

Figure 1 ECore meta-model of the ETLCL language.

Concrete syntax and language implementation

Developing a new DSL is a challenging task that involves defining syntax, creating syntax and semantic analyzers, and developing accompanying tools such as editors and debuggers. In recent decades, significant efforts have been made in the development of environments that facilitate language development by supporting automated generation of syntax analyzers and editors, as well as by providing languages for model transformations. One such platform, named the Eclipse Modeling Framework (EMF), was used to develop the ETLCL language. EMF provides the xtext framework that generates a syntax analyzer based on a grammar specification using this tool's grammar language; xtext also generates a basic editor that can be used within the Eclipse IDE, offering common functions such as syntax coloring and code completion.

The concrete syntax of the ETLCL language is textual. When selecting syntactic rules, notations were chosen that are familiar to users in the concrete DW domain. Our decisions regarding the selection of concepts are also guided by the need to fulfill requirement R1. The majority of ETL language commands are related to the creation of tasks, schedulers, and execution workflows. For these commands, syntax was used that resembles SQL statements. For example, the command for creating a task is: Group1.CREATE_TASK(TASK_UNIQUE_ID 1). ETLCL exploits the object notation, so a user may easily access a list of commands available for an object. Table 1 provides an overview of several grammatical rules related to the creation of tasks and related concepts, which are specified using xtext grammar language. The formal specification of the grammar will not be described in this article for the sake of readability, but the complete specification is available in the Supplemental File.

Table 1 The ETLCL grammar rules.

Create_environment:	
  name = ID '.CREATE_ENVIRONMENT' '('	
  ('ENVIRONMENT_DESCRIPTION:' desc = STRING)	
  ');'	
;	
Create_load:	
  env = [Create_environment] '.CREATE_LOAD' '('	
  ('LOAD_NAME:' name = ID)	
  ');'	
;	
Create_group:	
  load = [Create_load]'.CREATE_GROUP' '('	
  ('GROUP_NAME:' name = ID)	
  (',' domain = [Create_domain])?	
  ');'	
;	
Create_task:	
  group = [Create_group] '.CREATE_TASK' '('	
     ('TASK_UNIQUE_ID:' name = ID)','	
     ('TASK_NAME:' tname = STRING)','	
     ('AGENT_NAME:' aname = [Create_agent])	
     (',''PARAMETERS:' parameters += Parameters)*	
  ');'	

Code generation and run-time environment

An ETLCL specification is a starting point for the generation of various executable specifications, such as SQL code, ETL tools commands, and OS commands. The code generation is executed in the context of a selected platform. The xtend framework was used for model-to-code transformation.

The execution environment of ETLCL is based on the concept of a meta-data repository and services that operate on this repository. The first step in the code generation process is the generation of SQL code for creating a meta-data repository. The meta-data repository is a database containing specifications for ETLCL concepts such as tasks, groups, and dependencies. It is stored in a host’s DBMS, and the generated SQL code is adjusted for the target DBMS. In its present version, the ETLCL metadata repository consists of more than twenty tables. This approach is common in the field. For example, the ODI tool uses a host’s DBMS to store the whole working repository. In addition to the SQL statements for creating the metadata repository, statements are generated for populating tables within the repository, based on the specifications of tasks, groups, and dependencies. The metadata repository also serves as a working repository, with execution results and logs stored in specific tables designed for this purpose.

The ETLCL run-time environment, in addition to the meta-data repository, comprises several services that must be active to support the orchestration of tasks. This classification into four logical services simplifies difficulties caused by the complexity of the underlying system. The orchestration process is performed using the following services: Scheduling service

Manager service

Task service

Execution service (i.e., agent)

The scheduling service was implemented over the repository database. This service runs continuously and periodically queries the repository to create a list of loads ready for execution. If the list is not empty, then the manager service is called. The scheduling service execution algorithm is presented in Fig. 2.

Figure 2 Scheduling service execution.

The manager service coordinates the execution of the two remaining services: task service and execution service. The manager service updates the repository meta-data necessary for the execution of selected tasks and loads. Figure 3 illustrates the execution of the manager service. The first step is calling the task service, which analyses the meta-data repository, including the tasks, their execution statuses, and dependencies. The task service then creates a list of tasks ready for execution and the manager service initiates the execution of each task in the list. An agent designated for task execution runs the task and returns the status to the manager service.

Figure 3 Diagram of the manager service execution.

The task service reads the current information about tasks, analyzes dependencies, and prioritizes task executions based on this information, as well as domain identifiers and the values of appropriate parameters. It then creates a list of tasks ready to be executed by agents, which is returned to the manager service.

The execution service is active during the execution of the manager service. The execution service calls agents to run tasks that are ready for execution. If several independent tasks are ready for execution, these tasks will be run in parallel.

In the current version of ETLCL, these services were implemented for Microsoft SQL Server and Oracle DBMS for both Linux and Windows platforms. All services were implemented in a Java environment for easier portability to various operating systems. The execution service, via the agent concept, carries out tasks that represent atomic activities implemented using widely adopted commercial tools. At its current development stage, we have facilitated task execution within leading tools such as Informatica, Microsoft SSIS, ODI, and Talend. The choice of architecture and technological solutions for the execution environment aim to fulfill requirement R4. In this approach, an ETL expert, rather than orchestrating by writing a main package within a specific ETL tool and/or scripts for a particular OS and DBMS, utilizes the ETLCL language to specify the orchestration scenario for ETL activities. The orchestration itself is executed by the language’s execution environment, functioning as an abstraction layer above the specific computational environment.

Related work

In recent decades, ETL has been applied in various domains, including finance, healthcare, and telecommunications, attracting significant attention from the academic community and leading to numerous techniques, methodologies, and tools being developed to address various issues in this field. Nwokeji & Matovu (2021) provided an overview of research work in this area in their systematic literature review, including the main challenges of developing ETL solutions. They identified maintenance and the lack of automation as the primary challenges encountered in practice. One of the main objectives of ETLCL is to assist users in orchestrating and maintaining ETL processes. The most prevalent approaches to implementing ETL solutions are those based on conceptual modeling. The majority of these approaches are based on existing modeling techniques including UML diagrams, BPMN, and Semantic Web.

The idea of conceptual ETL modeling started with Vassiliadis, Simitsis & Skiadopoulos (2002) and their generic meta-model for ETL activities. Their proposal was to develop a methodology that would focus on the initial stages of data warehouse design. The aim was to analyze the structure and content of the available data sources and map them to the common data warehouse model.

Skoutas & Simitsis (2006) and Skoutas, Simitsis & Sellis (2009) outlined an ontology-based approach to assist in the conceptual design of an ETL process where datastores were conceptually represented by a graph and semantically annotated by a suitable ontology. The mappings between them could be subsequently inferred and automated reasoning was used to deduce correspondences and conflicts that may exist among the datastores.

Trujillo & Luján-Mora (2003) proposed the first UML-based approach, which involved conceptual modeling and specification of common operations in ETL processes. These operations included integration between data sources and transformation between source and target attributes. To facilitate the decomposition of ETL process design into logical units, the ETL process was constructed using UML packages, with each specific ETL mechanism represented by a stereotyped class. Since their research was early phase research, no code generation from specific models was provided. The next attempt at using UML (Muñoz et al., 2008) for ETL specification focused on Activity Diagrams (AD). Developers can readily design ETL processes at various levels of detail by employing stereotyped classes to represent each activity. However, this approach does not provide a way to represent time constraints, dynamic aspects, or the sequencing of control flows.

Another frequently used modeling technique is Business Process Model Notation (BPMN). BPMN offers a set of conceptual tools for graphical representation and specification of business processes that can be transformed into a target execution language. Additionally, it serves as a universal notation for designing all enterprise processes in a consistent manner, enabling seamless communication between different processes. El Akkaoui & Zimanyi (2009) proposed an approach for ETL specification based on BPMN. Their idea was to customize BPMN for designing ETL processes by identifying fundamental constructs grouped into four categories: flow objects, artifacts, connecting objects, and swimlanes. They also showed how the conceptual model could be translated into executable specifications using Business Process Execution Language (BPEL), a standard executable language for specifying interactions with web services.

Mazón et al. (2013) highlighted that ETL designers encounter two key challenges when developing ETL processes: (i) determining how to implement the designed processes in a target language, and (ii) maintaining the implementation when the organization’s data infrastructure undergoes changes. They proposed the BPMN4ETL—a model-driven development (MDD) framework based on BPMN for ETL processes. The framework used model-to-text transformation to assist ETL engineers in executing designed processes in an executable language and model-to-model transformation to ensure that the implementation updated as the target infrastructure changed over time.

Awiti, Vaisman & Zimanyi (2020) introduced an extension of relational algebra (RA) that incorporated update operations for defining ETL processes at a logical level. They addressed the case of slowly changing dimensions (SCDs) and compared this approach against BPMN4ETL and showed that the SQL implementation using RA to translate the specification into SQL runs significantly faster than BPMN4ETL.

Oliveira & Belo (2015) and Oliveira et al. (2019) presented frequent patterns in ETL processes using BPMN notation including surrogate key pipelining, slow-changing dimensions, and change data capture. Using the BPMN notation, they demonstrated how ETL patterns could be used to represent a typical ETL process, and how these patterns could be consolidated into a single ETL system package.

Biswas et al. (2019) proposed another modeling approach for conceptualizing the ETL process using a standard systems modeling language (SysML), which is a new modeling language standardized by the Object Management Group (OMG). Their article additionally focused on creating an automated executable SysML model based on an activity diagram. The aim of their study was to bridge the divide between modeling and simulation, and to assess the effectiveness of the proposed SysML model by evaluating its performance.

A technology-specific modeling method for the management of ETL processes within an organization was presented by Deme & Buchmann (2021). They used AMME methodology to design a modeling method for model-driven ETL process execution. Their article also reflected on a flavor of conceptual modeling languages labeled here as “technology-specific” distinguished from the traditional class of “domain-specific” languages.

Most frameworks typically focus either on DW design or on ETL process modelling. However, Atigui et al. (2012) proposed a generic unified method that automatically integrated DW and ETL design. Their approach employed the MDA framework and used UML profile and diagrams for DW and ETL design, while the extraction formulas were formalized using an OCL extension.

Song, Yan & Yang (2009) designed an ETL metamodel using the UML profile where ETL processes are decomposed into a set of ETL operations, such as merge, join, and filter, to decrease their complexity. These operations are then used as a starting point to create UML profiles for ETL design at a conceptual level.

In their work, Albrecht & Naumann (2013) emphasized that within a typical production environment, numerous individual ETL workflows evolve continuously as new data sources and requirements are integrated into the system. Managing these often intricate ETL workflows can be a challenging endeavor. To address this issue, the authors proposed an ETL management framework that supports the development and maintenance of ETL workflows through high-level operations like searching, matching, or merging entire ETL workflows. Users can then use these operators to search through a repository containing ETL workflows and, if necessary, perform merging to optimize the execution of ETL activities.

Approaches to designing a data warehouse (DW) often assume that its structure remains constant. In reality, the structure of a DW undergoes many changes, primarily due to the evolution of external data sources and alterations reflecting real-world dynamics within the DW (Wrembel, 2009). Wojciechowski (2013) suggests that changes in external data sources play an important role in data warehousing systems. These changes can involve both content and structural modifications. Structural changes often result in errors, necessitating the redesign and redeployment of an ETL workflow after each change. Frequent manual modifications of an ETL workflow are complex, error-prone, and time-consuming. To mitigate this problem, Wojciechowski introduced an environment, called E-ETL, for detecting structural changes in external data sources and managing these changes within the ETL process.

The majority of the literature on ETL process modeling concentrates on the utilization of widely-used, “general-purpose” modeling languages, such as UML and BPMN. These modeling languages are often used to reduce ETL development and maintenance costs. Most published approaches are also process-centric and focus on the initial phase of the ETL process design, such as ETL flow description and specification of typical tasks. Most proposed solutions do not cover later development and deployment stages of ETL processes, such as maintenance and orchestration, as it is a case in ETLCL.

ETLCL considers documentation and maintenance of existing ETL processes in its organization and leverages the reusability of common tasks. Using ETLCL, ETL designers can browse repositories of tasks and flows, update flow definitions, or create new flows by reusing existing ones. Furthermore, a new DSL proposed in this article introduces a set of concepts to describe ETL tasks and their relationships instead of using semantically rich languages, such as BPMN, that are capable of modeling an arbitrary process. Therefore, ETLCL’s unique contributions to the existing body of literature are the following capabilities: Specifying and managing existing ETL processes deployed in different execution environments;

Maintaining a metadata repository of existing tasks, flow, and configurations across various execution platforms; and

A minimal but sufficient number of concepts to semantically represent elements in ETL processes.

Another important characteristic that differentiates ETLCL from other attempts to conceptualize ETL modeling is its ability to orchestrate and maintain ETL processes across various execution environments.

Main concepts of the ETLCL language

ETLCL is a DSL aimed at controlling and maintaining ETL processes in DW. Users can orchestrate ETL processes using platform-independent concepts and commands that are translated into specific commands for a selected ETL tool and platform. ETLCL abstracts concepts that are inherent to most ETL tools. The ETLCL language meta-model is presented in Fig. 1.

The central concept of the ETLCL language is a Task. A task represents an atomic ETL activity. Various constraints over the execution of tasks are specified using the dependency concept. For example, dependency is used to establish a constraint so a task can only be executed if the previous task has been completed successfully. Tasks are organized into groups and each group can include various dependencies.

Configuration

The configuration concept is used to specify the target platform for which the program code will be generated and define the environment over which ETL orchestration is performed. The configuration concept is used to specify the ETL tool, the operating system on which the ETL tool is installed, the DBMS, and authentication parameters. These specifications are used by the code generator when generating SQL statements for creating the metadata repository. These specifications are also used as a starting point for generating procedures and functions that implement services for ETL process execution and management. An example configuration for SSIS/Windows and ODI/Linux platforms is presented in the beginning of the ETLCL code in Tables 2 and 3.

Table 2 ETLCL specification of an SSID ETL process.

TEST_CONFIGURATION1.CREATE_CONFIGURATION(	
   ETL_TOOL:SSIS,	
   ETL_TOOL_V:"12.0",	
   ETL_SERVER_ADDRESS: "localhost",	
   ETL_SERVER_OS: WINDOWS,	
   DWH_DB: MSSQL,	
   DWH_DB_V: "12.0",	
   DWH_DB_ADDRESS: "localhost",	
   DWH_DB_OS: WINDOWS,	
   DWH_DB_AUTH: "Windows",	
   ENVIRONMENT: "TEST Architecture",	
   ENVIRONMENT_DESC: "Test architecture for ETLCL language"	
);	
MSSQL_job.CREATE_SCHEDULER_TYPE(	
   REFRESH_TIME_IN_SEC:10,	
   LOG_DETAIL_F: 1	
);	
TestEnv.CREATE_ENVIRONMENT(	
   ENVIRONMENT_DESCRIPTION:"Test"	
);	
Test_context.CREATE_CONTEXT(	
   DESCRIPTION:'Test context,	
   ENVIRONMENT_NAME: TestEnv	
);	
TestLoad .CREATE_LOAD(	
   LOAD_NAME: Test_load	
);	
MSSQL_PROCEDURE.CREATE_AGENT_TYPE(	
   DESCRIPTION:"MSSQL PROCEDURE agent type"	
);	
SQL_PROCEDURE.CREATE_AGENT(	
   AGENT_TYPE_NAME: MSSQL_PROCEDURE,	
   ENVIRONMENT_NAME: TestEnv	
);	
Test _load.CREATE_GROUP(	
   GROUP_NAME: Group1	
);	
Gr oup 1.CREATE_TASK(	
   TASK_UNIQUE_ID: T1,	
   TASK_NAME: "Package1.dtsx",	
   AGENT_NAME: SQL_PROCEDURE	
);	
Gr oup 1.CREATE_TASK(	
   TASK_UNIQUE_ID: T2,	
   TASK_NAME: "Package2.dtsx",	
   AGENT_NAME: SQL_PROCEDURE	
);	
Gr oup 1.CREATE_TASK(	
   TASK_UNIQUE_ID: T3,	
   TASK_NAME: "Package3.dtsx",	
   AGENT_NAME: SQL_PROCEDURE	
);	
T 1 .DEPENDS_ON_TASK. T 2 (SUCCESS);	
T 2 .DEPENDS_ON_TASK. T 3 (ERROR);	
MY_SCHEDULER.CREATE_SCHEDULER(	
   LOAD_NAME: Test_load,	
   CONTEXT_NAME: Test_context,	
   DATE_TIME: "01.01.1970 00:00:00",	
   DESCRIPTION: "Scheduler Service"	
);	

Table 3 ETLCL specification of an ODI ETL process.

TEST_CONFIGURATION1.CREATE_CONFIGURATION(	
   ETL_TOOL:ODI,	
   ETL_TOOL_V:"12c",	
   ETL_SERVER_ADDRESS: "localhost",	
   ETL_SERVER_OS: LINUX,	
   DWH_DB: ORACLE,	
   DWH_DB_V: "19.1.0",	
   DWH_DB_ADDRESS: "localhost",	
   DWH_DB_OS: LINUX,	
   DWH_DB_AUTH: "username:***,pass:**",	
   ENVIRONMENT: "TEST Architecture",	
   ENVIRONMENT_DESC: "Test architecture for ETLCL language"	
);	
….	
Grupa1.CREATE_TASK(	
   TASK_UNIQUE_ID: T1,	
   TASK_NAME: "TABLESPACE_CHECK 001",	
   AGENT_NAME: ORACLE_PROCEDURE,	
   PARAMETERS: INSTANCE: "agent_jco_npc",	
);	
….	

Task

A task is the atomic execution unit of an ETL process. In code generation, this concept is translated into a package in the SSIS tool, a workflow in Informatica, or a scenario in the ODI tool. It can also be translated into a job or stored procedure over a DBMS. Task parameters are used to define the following: name of a procedure, package, or job that executes business logic associated with the task,

task identifier,

domain identifier, and

user-defined parameters.

The ETLCL code is provided for the examples in Tables 2 and 3, which includes the specifications for tasks when they represent packages within the SSIS tool, as well as scenarios within the ODI tool.

Dependency

In many cases, an ETL process depends on the execution status of another process. Dependencies are used to specify such constraints. Dependencies are also used to determine the order of task execution and avoid concurrent access to a table. There are three classifications of dependencies among tasks: On Success—the dependent task will be started only if the preceding task is completed successfully,

On Completion—the dependent task will be started only if the preceding task is completed, regardless of its status, and

On Error—the dependent task will be started only if the preceding task is completed unsuccessfully.

These three types of dependencies can also be established among groups. The dependency between two tasks or two groups is defined using the commands DEPENDS_ON_TASK and DEPENDS_ON_GROUP.

Group

The group concept is a logic unit for grouping tasks. A group may contain an arbitrary number of tasks with associated dependencies. Each group must have both a unique name and a domain identifier.

Load

The load concept represents the highest level of grouping. A load is used for modeling the overall process of transferring data from an origin to a destination. Each load is uniquely identified by its name and associated with an environment. The date and time of execution, as well as the execution periodicity, are defined using the scheduler concept.

Domain

A domain also models constraints among processes. In practice, there may be situations where certain constraints cannot be effectively expressed using dependencies. For example, two separate tasks might simultaneously use a table, with one task writing data to the table and the other task reading data from the table. The domain allows users to mark the tasks or groups that share the same resources. Tasks and groups with the same domain identifier wait for another task or group using the same identifier to complete its execution before starting to ensure they are executed in the correct order and do not overlap or interfere with one another.

Scheduler

A scheduler is a concept that defines the order in which loads should be executed in an ETL process. A user may define the start time of a load using an appropriate scheduler property. Scheduler is also used to define an interval for tasks that require iterative execution.

Environment and context

The environment and context concepts represent a set of software and hardware components where the load will take place. These concepts are used to specify all the loads that will execute in a particular environment and context.

Agent

The agent is the main concept for specifying how tasks will be executed. An agent specification is translated into a server script containing commands for running processes within an ETL tool. An agent executable specification includes commands for running the associated task, writing log files, modifying data, and returning results.

Each agent type has a set of parameters. The agent type determines an execution context, such as an ETL tool or operating system. Table 4 shows all supported agent types and parameters.

Table 4 Supported agent types.

Agent type	Parameters	
Informatica—Linux	Server, information system name, working directory, workflow, and session name.	
Informatica—Win	Server, information system name, working directory, workflow, and session name.	
SSIS—File system	File name, execution mode (×86/×64).	
SSIS—SQL Server	Package name, SQL server address, username and password, execution mode (×86/×64).	
SSIS—SSIS Package store	Package name, SQL server address (Widows authorization is used), execution mode (×86/×64).	
SSIS—Integration Service Catalog	Directory, project name, package name, execution mode (x86/x64).	
ODI—agent	Instance name, scenario name, scenario version, agent name, agent location, and execution context.	
Talend	Script file path, context name.	

ETLCTL specifications of a selected ETL process

This section presents a concrete example of ETLCL specifications for a typical workflow and demonstrates how these specifications are interpreted for two different ETL tools and two distinct platforms. Table 2 contains an example specification for loading data into a DW. The CREATE_CONFIGURATION command is used to specify that the target platform consists of the SSIS tool and MSSQL DBMS installed on the Windows operating system. Environment, load, and group are created using the CREATE_ENVIRONMENT, CREATE_LOAD, and CREATE_GROUP commands. In this example, there is one load named Test_load that contains one group of tasks named Group1. The group consist of three tasks identified as T1, T2, and T3, representing the Package1.dtsx, Package2.dtsx, and Package3.dtsx packages in the SSIS tool. The DEPENDS_ON_TASK command is then used to specify that task T2 will only be executed if task T1 successfully completes its work, while task T3 will be executed only if T2 completes its work with a status indicating an error.

In this specific environment, the scheduling service is triggered every 10 seconds, as defined by the CREATE_SCHEDULER_TYPE command and the REFRESH_TIME_IN_SEC parameter. The CREATE_SCHEDULER command creates a scheduler that will initiate the load on a specific date and time. Task T1, i.e., Package1.dtx, is the first in the group to be ready for execution since it does not depend on the execution of other tasks, so the corresponding agent is invoked. For this environment, the agent retrieves information from the SSIDB catalog and initiates the appropriate OS command, which is executed using the xp_cmdshell command. This command returns the execution status of the package, and this status is saved into the appropriate log table, which is part of the metadata repository. The execution status is then used to determine whether to execute Package2.dtsx. In a similar manner, when the execution of this package is completed, the return status value is read, and a decision is made on whether to execute Package3.dtsx.

A second example specification, outlined in Table 3, used the same load scenario but on a different target platform, which consisted of the ODI tool and ORACLE DBMS installed on the Linux operating system. Table 3 contains the ETL specification of the same scenario for this platform, excluding the environment, load, groups, and dependencies, as those remained the same as in the previous example. For this platform, tasks represent scenarios in the ODI tool. Based on the specification of task T1, the ODI scenario TABLESPACE_CHECK 001 is executed. When the load is ready for execution, just as in the previous example, task T1 will be the first one ready for execution. The task execution service calls an agent for the ODI tool, which triggers the following Linux command: ./startscen.sh-INSTANCE=agent_jco_npc TABLESPACE_CHECK 001 PROD. The execution results and all logs can be read from the ODI repository. For every scenario executed by the agent, information is logged about the time, execution success, execution method, and the call details. These logs are then used to determine whether tasks T2 or T3 will be triggered.

Application of etlcl in a use case

The common approach for orchestrating a large number of ETL workflows in DW systems involves creating a central ETL package within a specific ETL tool. This master ETL package serves as the core hub, calling all other packages, DW procedures, functions, and related elements. Within this master ETL package, all dependencies are defined, along with the necessary variables required for metadata transfer and execution. This parameterization approach offers several advantages, such as facilitating ETL load upgrades during the development of new mappings and table loading. Furthermore, centralization enables users to more quickly gain a comprehensive understanding of the data transfer logic from source to destination. However, this centralized approach has limitations when it comes to task manipulation. For instance, removing a single link representing a dependency can sometimes lead to intricate package reorganization. Any upgrade to the main package, such as adding a new task, commonly demands a more in-depth understanding of the ETL tool itself.

Several representative use-cases are presented in this section to illustrate the advantages of ETLCL for orchestrating ETL processes in DW systems compared to the approaches that are used in practice today. The selected use-case scenarios cover the majority of patterns that exist in practice today, grouped as follows: (i) changing the logic of the load process, (ii) changing the DW architecture, and (iii) switching to a new ETL tool. The first two groups include scenarios that are frequently encountered in DW systems and refer to situations such as adding a new task or introducing a new data mart. Six scenarios are included, covering the most common situation encountered in DW systems. Switching to a new ETL tool is less common, but can be a very demanding process, so it was also included to demonstrate how ETLCL can help facilitate the process.

Table 5 shows the estimated time required for all use-case scenarios included, with time comparisons between a widely-used commercial ETL tool and the ETLCL language. These estimations are based on previous repetitive executions of these scenarios in real-world systems, performed by the ETL process designers of an equal experience and level of knowledge in the same software company.

Table 5 Estimation of time needed to execute the following scenarios.

Scenario	Estimated time using an ETL tool	Estimated time using ETLCL	
1. (a)	8 man-hours	1 man-hour	
1. (b)	16 man-hours	2 man-hours	
1. (c)	16 man-hours	2 man-hours	
1. (d)	This scenario is not usually supported by ETL tools	Performed by the engine for an very short amount of time	
1. (e)	On demand—it depends on the operation team	10 man—minutes	
1. (f)	1 man-hour	10 man—minutes	
2.	No more than two man-days	No more than 2 man-hours	
3.	N/A	No more than 3 man-hours	

Changing the logic of the load process

Changing the logic of the load process is very common in DW systems that are constantly upgraded according to user needs, including adding new tasks and jobs, changing dependencies between two or more processes during data transfer, or changing the error processing logic. All these changes have to be implemented through an ETL tool or at the procedural level. ETLCL, however, enables the specification of changes at a higher level of abstraction, while the engine in the background transforms these specifications into a new implementation of the existing tasks. This approach simplifies the implementation of the changes and shortens the time required to perform them.

In the following scenario, the change of load logic is compared using ETLCL and the SSIS tool, which is a popular, concrete ETL tool. In general, ETL/ELT tools share many common concepts, as described previously. SSIS does not have a tool to define the order between job calls. When an additional job needs to be created, the main job must be manually modified, as it contains the logic of start-up dependencies, error processing, calling services, and sending notifications of all jobs. Table 6 illustrates the process of creating a new job using ETLCL. ETLCL provides commands to include a newly created job into the existing load. First, a new SSIS package, or job, named Package_2.dtsx, is created in the listing. Then, a new task is defined and included in an appropriate group. ETLCL commands are used to define dependencies and error handling. The SSIS also does not provide automatic restart functionality, unless it is implemented using markers and dependencies. The ETLCL environment stores all metadata into the database, so restart functionality is supported by the engine.

Table 6 Adding a new task to a group.

Group1.CREATE_TASK(	
   TASK_UNIQUE_ID: Package_2,	
   TASK_NAME: "Package_2.dtsx",	
   AGENT_NAME: SQL_PROCEDURE	
);	
Package_2.DEPENDS_ON_TASK.Package_1(SUCCESS);	
PackageStandardError.DEPENDS_ON_TASK.Package_2(ERROR);	

Changing the DW architecture

Changing the DW architecture usually implies the introduction of a new data mart, layer in the architecture, or modification of task schedule and dependencies. These changes require significant effort when load logic is manually defined.

When adding a new data mart whose dimension tables are already included in the main DW environment, creating a dependency between the existing environment and the new data mart requires defining a new and more complex structure of the main package. Using ETLCL, this can be done by simply creating dependencies with the DEPENDS_ON_TASK command, then the new data mart load will wait for the last successful load of all related tasks.

Switching to a new ETL tool

Switching to a new data loading concept or ETL tool is an extremely demanding process that involves rewriting the specification of existing tasks. It also requires the creation of a new package or framework that will implement the logic of dependencies, errors, restart options, etc. In ETLCL, the existing settings can be used with a set a new configurations for the tool.

Other use-case scenarios of changes that occur frequently in DW systems:

1) The following scenarios relate to the modification of the existing load logic, such as adding a new task, that require changing the settings of the input and output parameters, the method and period of the task launch, as well as the dependencies of the existing tasks: Adding a new task to the existing daily load: When this is done on a concrete ETL tool, such as Oracle Data Integrator, the connection on dependencies must first be removed, a new task inserted, and the parameters set for the start mode (parallel or sequential). The branch must be defined after successful execution and in case of an error, and the input parameters set. These steps must be performed over the development environment, regenerating the main package, refreshing the launch agents and their schedule, performing testing, and then redeploying to the production environment. Because of the number of steps required and their complexity, an ETL expert is required. With ETLCL, these steps may be performed with ETLCL language commands, including creating a new task with the package name, adding a dependency over existing data, and adding the task to the group from which the start-up time and error handling are inherited.

Changing the definition of a load in a way that requires including a new database procedure or script: This scenario is more complex than the previous one since each ETL has its own mechanisms for invoking a procedure or script within the main package. Also, some ETL tools do not provide support for invoking such external scripts, requiring a scheduler to be implemented at the operating system level. The steps for adding a procedure or script within an existing package using a concrete ETL tool are identical to the steps described in the previous scenario, but additional settings need to be specified to determine the way and time external scripts and procedures are triggered. With ETLCL, the set and order of commands to be executed can be defined using language commands, as in the previous scenario.

Setting multiple dependencies for a new task: Multiple dependencies means tasks need to wait for the successful execution of a mapping, script, procedure, or process outside of the ETL tool itself. In most cases, this requires fairly complex steps that can take a lot of time during implementation and testing. ETLCL, however, provides appropriate commands for multiple dependencies, so this type of dependency setting does not require additional steps.

Creating a restart point: The ETLCL language provides commands for monitoring task execution and logging, enabling quick and easy recovery from an error and starting the next load process from the point of its interruption.

Manual initiation of individual tasks is also common in DW systems. In practice, users often need to start a task or a group of tasks on demand, and such a request must be forwarded to the developer for execution. On the other hand, ETLCL provides appropriate commands for such an operation.

2) Adding new sources, or destinations, and changing the DW architecture usually requires significant effort in changing configuration parameters, reorganizing the main package, testing the dependencies, and redistributing the tasks. Using the ETLCL language, configuration changes are performed in a centralized way within the configuration table, significantly reducing the time it takes to insert new tasks or a new group of tasks.

3) Changing the version of the ETL tool, or switching from one ETL tool to another requires numerous changes for implementation, such as changing the dependencies, the way tasks are started, and processing errors. Using ETLCL, these changes require significantly less effort since the engine is able to translate the existing dependencies in accordance with the new ETL tool.

An assessment of etlcl’s main characteristics

Several studies have focused on the theoretical aspects of DSL development (Mernik, Heering & Sloane, 2005; Kosar et al., 2010; Lukovic et al., 2012; Kahraman & Bilgen, 2013). This section covers ETLCL quality characteristics as presented in Kahraman & Bilgen (2013). Analyzing our language from the perspective of end-users was one of our main goals. However, we were unable to conduct formal interviews using questionnaires to gather feedback on our language because we do not have enough trained users to provide statistically valid results. On the other hand, we have included potential users in the implementation of test cases, allowing us to gather their feedback and present some of their testimonials. We gave particular focus to productivity and portability, which are the primary objectives of ETLCL, among other factors. In order to analyze the key characteristics of our DSL, we tested it using the scenarios that were described in the previous section.

Productivity of a DSL refers to the degree to which a language promotes programming productivity. Productivity is a characteristic related to the number of resources expended by the user to achieve specified goals (Kahraman & Bilgen, 2013). To assess the productivity of the ETLCL language and verify the satisfaction of requirement R5, a small study was conducted at Logate ltd., which is one of the largest software companies in Montenegro. Four professionals in the field of ETL with many years of experience executing these scenarios in real-world systems were interviewed. The second column of Table 5 provides estimations of the average time required to execute each scenario using widely-used commercial ETL tools. A short training on using the ETLCL language was then conducted, where users were introduced to the main concepts and commands of this language. The estimated time required to execute each scenario using ETLCL is listed in the third column of Table 5.

To assess the usability of ETLCL, the ETL experts were asked to provide feedback specific to ETLCL’s usability in informal interviews, including one strength and one area needing improvement. In general, the assessment of the ETLCL’s usability was positive. The participants highlighted that the syntax of the language was comparatively easy to learn, as it incorporates familiar keywords and commands commonly used ETL languages and tools. From this, it can inferred that requirement R1 has been largely fulfilled. However, ETLCL’s lack of advanced tooling, especially including a visual editor, was emphasized as a significant shortcoming.

Requirement R3 is closely tied to functional suitability that refers to the degree to which a DSL is fully developed. This means that all necessary functionality is present in the DSL (Kahraman & Bilgen, 2013). ETLCL has a high degree of functional suitability and meets the requirement R3 in the domain of ETL orchestration as all test case scenarios were able to be described using this language.

According to Kahraman & Bilgen (2013), extensibility refers to a language’s ability to incorporate new features, while integrability measures how easily a DSL can integrate with other languages and modeling tools. ETLCL exhibits a low degree of both extensibility and integrability as it lacks mechanisms for incorporating new functionalities and integrating with other languages.

The ETLCL language facilitates portability. Defining and maintaining ETL processes is a highly error-prone task, even for experienced, skilled professionals. Using ETLCL as a PIM language that provides high-level concepts and commands mitigates the risk of error. With ETLCL, introducing a new platform, target language, or ETL tool does not require creating new specifications for ETL processes from scratch. Instead, it only requires amendments of existing transformation algorithms. Therefore, it can be concluded that requirement R4 has been met in a large extent.

Conclusions

The ETLCL language is designed for software companies specialized in the design and implementation of ETL processes. ETL process designers and data engineers at these companies deal with numerous ETL processes using heterogeneous technological platforms. The ETLCL language helps designers specify and maintain complex ETL processes. This DSL solution makes several common ETL processes easier, including: Providing a high-level description of existing ETL processes;

Modifying ETL processes, including actions such as changing a sequence of tasks, adding new tasks, and removing existing ones;

Creating new ETL processes in an easy and consistent way;

Deploying and executing ETL processes specified by ETLCL; and

Reducing efforts when switching to a new ETL tool or technology platform.

Several directions for future work were identified during the development of the ETLCL language. These include enhancing the execution environment in order to support ETL tools in addition to the ones listed in Table 1. Future research will focus on improving the dependency specification and developing a system for automatically identifying conflicts in data loading using metadata and log files from ETL tools. Creating a graphical syntax for defining dependencies would also be beneficial, as it would make it easier to understand and manage dependencies. A formal evaluation of the ETLCL tool is also planned by conducting an experiment that will include individuals with previous experience in the DW domain, as well as those who may potentially use such a language. Results will be gathered in the form of questionnaires and then statistically analyzed.

Supplemental Information

Supplemental Information 1 Source code of the ETLCL language implementation.

The source code archive contains all the necessary files to implement the language, including (i) the EtlCl.ecore file containing the ETLCL language’s Ecore metamodel; (ii) the EtlCl.xtext file containing the specification of the language’s concrete syntax; (iii) the EtlClGenerator.xtend file containing the specification of the M2T transformations; (iv) the src-gen folder containing the classes implementing the language’s lexical and syntax analyzers; and (v) the xtend-gen folder containing the classes implementing the M2T transformations.

Additional Information and Declarations

Competing Interests

Author Contributions

Data Availability

The authors declare that they have no competing interests. Nikola Trajković is employed by Softwecs LTD, and he declares no competing interests.

Aleksandar Popović analyzed the data, performed the computation work, prepared figures and/or tables, and approved the final draft.

Vladimir Ivković analyzed the data, authored or reviewed drafts of the article, peformed reseach for related work, and approved the final draft.

Nikola Trajković performed the experiments, performed the computation work, authored or reviewed drafts of the article, and approved the final draft.

Ivan Luković conceived and designed the experiments, prepared figures and/or tables, authored or reviewed drafts of the article, and approved the final draft.

The following information was supplied regarding data availability:

The source code is available in the Supplemental File.

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
