# Peer review of "A domain-specific language for managing ETL processes"

_PeerJ Computer Science, doi:10.7717/peerj-cs.1835_

## Round 0.1 · original submission · Major Revisions

While the reviewers acknowledge the relevance of your manuscript, they recommend major revisions. It is essential to carefully consider and address all of their suggestions, with particular attention to incorporating any overlooked elements that are essential for contextualizing your contribution within the appropriate methodological framework.

**Language Note:** PeerJ staff have identified that the English language needs to be improved. When you prepare your next revision, please either (i) have a colleague who is proficient in English and familiar with the subject matter review your manuscript, or (ii) contact a professional editing service to review your manuscript. PeerJ can provide language editing services - you can contact us at copyediting@peerj.com for pricing (be sure to provide your manuscript number and title). – PeerJ Staff

Reviewer 1 ·

Basic reporting

The paper introduces a DS(M)L for ETL processes to support maintenance of data warehouses.

The problem is convincingly introduced and the solution tries to mitigate an inherent technological heterogeneity in data warehouse management.

Related works show good coverage with respect to ETL process languages but should also include considerations on DSML engineering methodologies and the research methodology actually leading to this DSL.

Even if the work itself entirely neglected such methodologies, it should describe the methodology that was actually used/improvised and it should try to frame it within the context of modeling language/method engineering methodologies that exist for more than two decades. Such methodologies will also provide a bit of structure to how such a DSL should be presented, for example:
- Requirements (which must be later revisited in the Evaluation phase, to show the ability of the language to satisfy them)
- Design decisions (metamodel, notation, semantics, syntactic constraints)
- Model-driven mechanisms (how is the code generation achieved, or other model-driven features, is there any design-time analysis mechanisms on top besides the run-time execution?)
- Model-driven environment (how does the language interoperate with the environment upon which it acts)
- Operating procedure (how is the model supposed to be used in a step-wise manner by those who adopt it)
- Evaluation (what are the qualities targeted by the language, how are those qualities measured, what is the competence of the language relative to the requirements that motivated its creation)

Some of these aspects are partly covered but spread around, disjointed from each other (e.g. difficult to connect the evaluation to requirements) or simply obscured in a structure that is rather fuzzy ("main concepts", "application", "discussion on characteristics"). This weak structure tends to blur distinctions and obscure a lot of relevant aspects, giving strictly an engineering view and less of a meta-level scientific view. The work should clearly distinguish building blocks of the modeling method emerging from this DSL (one idea would be to use modeling method building blocks as defined by Karagiannis: notation, syntax, semantics, modeling procedure, mechanisms supporting design-time, mechanisms supporting run-time).

Overall this is useful work that will be beneficial for both the journal and for the data management community. However the structure and style of reporting is really focused on the contribution and fails to frame it in the methodological paradigm where it belongs (its only framing is engineering-oriented, but for a journal publication I find that insufficient)

Experimental design

Generally, the paper makes a good job in indicating the gap that it addresses and describing the technical details on achieving that.

However the paper lacks entirely a methodology section (research&engineering methodology, not only usage), as it jumps directly into the proposed artifact and its applicability. Metamodeling as enabler for the engineering of DSMLs have a well established scientific background with different flavours proposed by different research groups:
- https://link.springer.com/chapter/10.1007/978-3-642-36654-3_6
- https://link.springer.com/chapter/10.1007/978-3-030-93547-4_1
- https://ieeexplore.ieee.org/document/8904580
- https://link.springer.com/book/10.1007/978-3-642-41467-1

Validity of the findings

Overall the presented use cases make a positive impression on how the DSL can be used and what its first-class constructs contribute as mediators for DHW management. Perhaps the title of the quality evaluation sections should make it clear that it is about evaluation ("discussion on characteristics" sounds too vague).

Cite this review as

Reviewer 2 ·

Basic reporting

The authors propose the ETLCL language which aims for interoperability between several ETL tools. ETLCL language allow users and companies who have BI systems and/or who use ETL tools to change from one tool to another or to use several at the same time without having to worry about interoperability and communication problems between their different tools. This language facilitates portability.

Certainly, the ETL tools used today do not communicate with each other, for this changing the tool used within a company is expensive and complex. The idea of providing a meta-model that covers the use of several tools is particularly interesting for companies forced to use an exclusive tool due to the lack of interoperability and portability.

The article is mainly easy to follow and understand. But unfortunately, there are several shortcomings that deserve to be addressed.

It is very unfortunate that the meta-model presented in Figure 1 is unreadable even with a zoom. Unfortunately, it is impossible for me to evaluate the different concepts of the meta-model which is very important to evaluate the quality of the contribution in this paper.

Experimental design

The study of the state of the art must be deepened. The contribution deserves more effort in formalization and definition as follows.

- The study of the state of the art must be deepened by adding for instance:
o Design ETL Metamodel Based on UML Profile, Xudong Song; Xiaolan Yan; Liguo Yang, 2009
o Using OCL for Automatically Producing Multidimensional Models and ETL Processes, Faten Atigui, Franck Ravat, Olivier Teste, Gilles Zurfluh: DaWaK 2012.
o The studies proposed in: « Progressive Growth of ETL Tools: A Literature Review of Past to Equip Future (Monika Patel & Dhiren B. Patel, 2020) » and « A Systematic Literature Review on Big Data Extraction, Transformation and Loading (ETL) Joshua C. Nwokeji & Richard Matovu, 2021 » may be useful to clearly define the concepts of the metamodel and defend the authors' contribution and justify its usefulness.
o and probably others existing studies. ETL process modeling has been widely studied by researchers and practitioners.

- I recommend listing the different concepts of the proposed DSL and their relationship with the ETL components of at least 2 ETL tools
- Also, I recommend adding a formal description of the meta-model with concrete examples of each concept (for example: Task; using SSIS task is... using Talend, etc.)
- In line 233: the authors say that they propose a minimal and sufficient set of concepts: how is the coverage of all possible cases and the minimality condition are verified?

- In scenarios of common changes in BI systems:
o Line 462: “load logic”: the modifications often concern the transform more than the load. I recommend changing the sentence to “ETL logic” to be more generic.
o Line 478: scenario b touches on a known problem in the evolution of DW. I recommend to study existing work to enrich your contribution at this level: A Survey of Managing the Evolution of Data Warehouses 2009, Robert Wrembel; Schema Evolution in Data Warehouses Zohra Bellahsene 2002 etc.

Validity of the findings

- I recommend adding a case study and full experimentation throughout the article: Take data sources and run ETL on at least 2 ETL tools with and without ETLCL.

- The example of Load presented in Figure 2 is very simple and does not add much to understanding. I recommend the authors to add a concrete example of a case study.


- It will be good to evaluate the contribution: I recommend using the ETLCL language in a concrete context and measure the benefits brought to the company and/or to users

Additional comments

- Authors use DWH as an abbreviation for Data Warehouse, it is more common to use DW and not DWH.

- Pay attention to the format of references and errors, for example online reference 616, the order of authors is not correct: « Mazon JN, Zimanyi E, El Akkaoui Z, Trujillo J. 2013. A BPMN-based design and maintenance 617 framework for ETL processes in International Journal of Data Warehousing and Mining 46:72 618 DOI: 10.4018/jdwm.2013070103”.

Cite this review as

---

## Round 0.2 · Minor Revisions

The revised version of your manuscripts has been significantly improved. However, a few minor issues still need to be addressed, as indicated by reviewer 1. Please, carefully revise your manuscript to address all the pending issues. I hope you can complete the recommendation changes in a revision of your article.

Reviewer 1 ·

Basic reporting

The paper was extensively revised and is much better grounded. The methodological gap was filled with a dedicated section that covers a bit of background and also reveals the actual methodology employed by the work at hand.

Some rather minor issues should still be addressed:

Language requirements and functional/run-time requirements should be clearly distinguished. Actually there are more categories of "modeling method requirements" involved in any DSL that is used for more than simply representation (e.g. requirements pertaining to notation, to level of semantic specificity, to model-driven mechanisms, alignment with the run-time environment etc. see https://ieeexplore.ieee.org/document/8920624). We put in a language exactly those constructs that are needed for how the language is intended to be used - e.g. in this case to achieve interoperability in the targeted ETL environment, to support the "logical services" driven by the DSL, to support how we plan to evaluate the "DSL characteristics".

This is still rather disconnected. The "requirements and design" subsection seems to only refer to the language requirements, seems to be decoupled from the subsequent discussion on the DSL use cases. The statement "Requirements R1 and R2 are generic to almost every DSL" is mistaken - obviously R2 refers to the ETL for DW application area. Actually the list of requirements should be enriched as much as possible with highly specific characteristics that are later reflected/relevant for use and final assessment. Having excessively generic requirements ("language concepts must origininate fromm the application domain") does not help, they are too weak to be presented as motivation for the work, their fulfillment impossible to discuss in the Assessment section.

There are obviously more concrete requirements that have been addressed by what was achieved - e.g. in terms of model-driven artifacts/interoperability or the so-called "logical services" (there is almost nothing about those in the requirements-focused paragraphs; if there are no requirements for those features, why were they implemented?).

Moreover, there is the requirement "R3 - Language concepts must be abstract and independent of specific technological platforms" but the implementation seems to be particularly focusing on SQL, Oracle/ODI, relational databases in general. Is the language really decoupled from technology, or is this decoupling only partial? Where exactly is the abstraction boundary, how is the bridging achieved over that boundary? Which technology-specific constructs are delegated "outside" the DSL and which were kept as concepts of the DSL? What was the general design rationale for setting this boundary?

Minor remark: The reference Ulrich (2013) implies that Ulrich is the family name of the author. In reality it should be Frank (2013)

Experimental design

no comment, technically and experimentally the paper looks good

Validity of the findings

The final evaluation findings should be better anchored in motivating requirements - these are now more noticeable than in the initial version, but still partly disconnected from the advertised characteristics and benefits of the DSL, as they are described in the second part of the paper. Problem identification, and its drilling down into detailed requirements that inform the assessed characteristics, is a critical phase in design science research (where DSL engineering fundamentally belongs)

Cite this review as

---

## Round 0.3 · accepted · Accept

Thank you for your contribution to PeerJ Computer Science and for addressing all the reviewers' suggestions. We are satisfied with the revised version of your manuscript and it is now ready to be accepted. Congratulations!